# Methods of Inactivation of Highly Pathogenic Viruses for Molecular, Serology or Vaccine Development Purposes

**DOI:** 10.3390/pathogens11020271

**Published:** 2022-02-19

**Authors:** Simon Elveborg, Vanessa M. Monteil, Ali Mirazimi

**Affiliations:** 1Department of Clinical Microbiology, Uppsala University Hospital, 751 85 Uppsala, Sweden; simon.olov.elveborg@akademiska.se; 2Clinical Microbiology, Department of Medical Sciences, Uppsala University, 751 85 Uppsala, Sweden; 3Department of Laboratory Medicine, Karolinska Institutet, 141 52 Huddinge, Sweden; vanessa.monteil@ki.se; 4National Veterinary Institute, 751 89 Uppsala, Sweden

**Keywords:** chaotriopic salts, heat, psoralens, 1.5-iodonaphtyl azide, UV-radiation, gamma-irradiation, Beta-propiolactone, aldehydes, aromatic disulfides, hydrogen peroxide, detergent, immunoassay, SARS-CoV-2

## Abstract

The handling of highly pathogenic viruses, whether for diagnostic or research purposes, often requires an inactivation step. This article reviews available inactivation techniques published in peer-reviewed journals and their benefits and limitations in relation to the intended application. The bulk of highly pathogenic viruses are represented by enveloped RNA viruses belonging to the *Togaviridae*, *Flaviviridae*, *Filoviridae*, *Arenaviridae*, *Hantaviridae*, *Peribunyaviridae*, *Phenuiviridae*, *Nairoviridae* and *Orthomyxoviridae* families. Here, we summarize inactivation methods for these virus families that allow for subsequent molecular and serological analysis or vaccine development. The techniques identified here include: treatment with guanidium-based chaotropic salts, heat inactivation, photoactive compounds such as psoralens or 1.5-iodonaphtyl azide, detergents, fixing with aldehydes, UV-radiation, gamma irradiation, aromatic disulfides, beta-propiolacton and hydrogen peroxide. The combination of simple techniques such as heat or UV-radiation and detergents such as Tween-20, Triton X-100 or Sodium dodecyl sulfate are often sufficient for virus inactivation, but the efficiency may be affected by influencing factors including quantity of infectious particles, matrix constitution, pH, salt- and protein content. Residual infectivity of the inactivated virus could have disastrous consequences for both laboratory/healthcare personnel and patients. Therefore, the development of inactivation protocols requires careful considerations which we review here.

## 1. Introduction

The factors that influence successful inactivation are determined by the inherent characteristics of the virus, the environmental setting (including the access to equipment and resources) and the nature of the intended downstream application. While serological analysis and vaccine development research set high demands on the preservation of protein structure, molecular biology tests merely require virus genome integrity. The focus of this article is to present the available inactivation techniques for viruses belonging to *Togaviridae*, *Flaviviridae*, *Filoviridae*, *Arenaviridae*, *Hantaviridae*, *Peribunyaviridae*, *Phenuiviridae*, *Nairoviridae* and *Orthomyxoviridae* families. Techniques intended for molecular-, serological assessments and vaccine development research are discussed in separate sections (Table 1 and Appendix A).

## 2. Molecular Biology

### 2.1. Guanidium Based Chaotropic Salts

Guanidium thiocyanate (GTC) constitutes the main component of nucleic acid extraction kits. GTC-containing buffers are extremely caustic and can be used to destabilize the viral envelope and eliminate cellular nucleases, while maintaining the structure of RNA/DNA for subsequent molecular biology analysis in lower-levels containment facilities [12,33,34,69]. The most common inactivation methods for nucleic acids extraction are TRIzol LS reagent (Invitrogen) and AVL buffer (QIAGEN viral extraction kits), but several other buffer systems can be used, sometimes in combination with other chemical agents [12,32]. J. A. Blow et al., 2004, demonstrated that TRIzol LS and AVL Buffer effectively inactivate viruses belonging to different viral families: *Togaviridae* (Venezuelan equine encephalitis virus (VEEV), Western equine encephalitis virus (WEEV) and Eastern equine encephalitis virus (EEEV)), *Flaviviridae* (West Nile virus (WNV), all Dengue virus serotypes (DENV)), *Filoviridae* (Ebola virus (EBOV), *Marburgvirus* (MARV)) and *Phenuiviridae* (Rift Valley fever phlebovirus (RVFV) [12]. The dilution of reagent leading to inactivation of the virus was determined by comparison of the presence of plaques in the mock-treated control wells and the absences of plaques in reagent containing samples at a range of non-cytotoxic concentration of reagent. The results indicate that TRIzol LS reagent and AVL buffer inactivate all tested viruses effectively and allow genetic characterization and diagnostic assays to be carried out in lower-level containment facilities (2). Guanidine-based extraction kits are also commonly used in the diagnostics of *Arenaviridae* such as Lassa virus (LASV) [13].

The guanidium-based agents commercially available may include different ratios and concentrations of active ingredients that affect their performance. In addition, their efficiency may be influenced by the nature and concentration of the pathogen, the sample matrix and the concentration and contact time of the inactivating agent with the sample [26]. Some studies have found that, while using AVL buffer alone leads to a significant decrease in EBOV titer, the virus may not be completely inactivated [15,32,70]. The addition of ethanol is required [26]. However, automated extraction systems do not permit the addition of ethanol by machine outside biocontainment. J.E. Burton et al., 2017 proposed that the addition of another chemical, such as Triton X-100, added earlier in the process, might be a safer alternative [26]. This study demonstrates the complete inactivation of EBOV after exposure to a combination of AVL and 0.1% Triton X-100 after incubation for 10 min. The combination of Triton X-100 and AVL buffer did not affect the performance of downstream RT-qPCR and next generation sequencing applications [26].

Rosenstierne et al., 2016, developed another rapid and simple bedside method for EBOV inactivation [32]. The commercially available GTC-containing Magna Pure lysis/binding (MPLB) buffer from Roche is directly injected into an ordinary vacuum blood collection EDTA tube. The blood drained into the tube is immediately inactivated. EBOV RNA is stable in MPLB buffer blood collection tubes for more than 5 weeks independently of the temperature. The RNA can then be extracted using either Magna Pure RNA extraction kit or QIAamp viral RNA minikit in resource-poor settings or field laboratories. With this method, direct hands-on handling of the potentially highly contagious substance is avoided and rapid diagnosis is facilitated. This facilitates daily monitoring of the patient’s viral load, differential diagnostics and safe transport of samples to other laboratories. A limitation of GTC-containing buffers is that serological and immulogical assays cannot be performed due to the caustic nature of guanidinium thiocyanate (to proteins and other macromolecules. MPLB buffer contains a combination of Triton X-100 and GTC, while AVL buffer contains GTC but no Triton X-100 [32]. Consistent with the findings of Burton et al., [26], addition of Triton X-100 to GTC buffer was found to potentiate its effect. Furthermore, MPLB buffer was showed to be effective against Vaccinia virus and Cowpox virus [71].

All the above studies concur with PAHO/WHO guidelines 2015 for handling of Ebola and other BSL-4 pathogens [8]. The aim of these guidelines is to establish some rules allowing safe laboratory testing that may lead to etiological diagnosis, evaluation and follow-up of the patient. For molecular diagnosis, the inactivation process should ensure a total loss of infectivity, while maintaining the integrity of the nucleic acids.

SARS-CoV-2 is effectively inactivated after 5 min incubation of Trizol LS (1:4 ratio of virus:Trizol), or with a detergent, 0.5% sodium dodecyl sulfate (SDS) or with Triton X-100 incubated for 30 min at room temperature [72,73]. This may be convenient for downstream diagnostic using qRT-PCR.

### 2.2. Thermal Inactivation

Thermal inactivation has been used as a method for inactivation of viruses and other infectious agents for 250 years. High temperatures alter viral structure (envelope and capsid) and proteins responsible for recognition and binding to the host cell, leading to inhibition of infection [1]. The thermostability of viruses and the rate of inactivation is extremely variable, even between closely related viral species [13]. It is influenced by virus concentration, serum protein content, pH and density of the solution, as well as the presence of organic compounds and salts [2]. These factors each have implications when selecting the thermal inactivation conditions for unknown viruses [2]. Protocols for heat inactivation, therefore, must be carefully designed and species specific [6,74]. Viral genomic RNA is relatively heat stable, but is denatured at temperatures higher than 60 °C [2,3]. Thermal inactivation of LASV (*Arenaviridae*) and other viruses such as Ebola and Marburg viruses (*Filoviridae*), Alkhurma hemorrhagic fever virus (*Flaviviridae*) as well as Chikungunya and Ross River virus (*Togaviridae*), has been demonstrated after exposure to 60 °C heat for 60 min [2,3,6,13,74]. As a safety measure WHO does not recommend that heat inactivation is used as a singular method, but that it be combined with a denaturing solution. The use of lysis solutions composed of guanidine salts is the recommended method of choice for inactivation of enveloped RNA viruses intended for molecular analysis [8,14,15,26]. Coronaviruses are sensitive to heat and therefore the WHO recommendation of previous SARS-CoV specimens were heat inactivation at 56 °C for 15 to 45 min [75]. A subsequent study on SARS-CoV-2 demonstrated effective inactivation at 56 °C for 30 min and at 65 °C for 15 min [76]. Genomic stability including the ORF1a, Spike, N and E genes were preserved at both temperatures and at 56 °C sensitivity to RNA detection by qRT-PCR was preserved [76]. Conflicting studies claim that up to 75 °C is required for successful inactivation of SARS-CoV on clinical samples [72,77].

### 2.3. Psoralens

Psoralens such as Amotosalen (previously known as S-59) and 4′-aminomethyl-trioxsalen (AMT) have been shown to inactivate a broad range of pathogens through inhibition of transcription and replication of nucleic acids [22,33,38,39,40] (Table 1). AMT is inactive until it is exposed to long-wave UV light (UV-A). Photoactivation causes formation of covalent bonds between the drug and one or two strands of nucleic acid [38]. Inactivation of DNA viruses by derivatives of psoralen was demonstrated as early as 1965 by Musajo et al. [78] and in subsequent studies for both DNA and RNA viruses [22,33,38,39,40]. AMT has been shown to efficiently inactivate a diverse set of RNA viruses including enveloped and non-enveloped, spherical and filamentous, positive and negative sense RNA genomes from seven different families [33]. Nucleic acids were sufficiently preserved to allow for molecular and genomic analysis. Virus inactivation by psoralen is dose-dependent and affected by the viral titer. The dose of AMT and length of UV-A exposure time required appear to be family dependent and, in some cases, even members dependent (Table 2) [33]. The inactivation of viruses of the *Filoviridae* family, for instance, requires a significantly higher dose than other virus families. In one study, EBOV and MARV were completely inactivated after exposure to 20 µg/mL of AMT and UV-A energy level of 1000 µW/cm^2^ for 120 and 150 min respectively, with total energy exposures of 120,000 and 150,000 µW/cm^2^ [33]. In comparison, VEEV (*Togaviridae*) and WNV (*Flaviviridae*) were inactivated after a total energy exposure of 2000 µW/cm^2^. All the tested *Flaviviridae* (DENV, WNV, SLEV and YFV) required a considerably lower dose (over 100 times) than *Filoviridae* for inactivation [33].

Five RT-qPCR assays designed for DENV genomes were successful following AMT-UV-A inactivation, indicating that the photoadducts created in the viral genome did not inhibit reverse transcription [33]. Moreover, flow cytometry showed DENV was internalized into cells, but was unable to replicate [33]. RT-qPCR was also successfully performed for *Arenaviridae* as well as RVFV (*Phenuiviridae*) and EBOV and MARV (*Filoviridae*), showing that RNA was sufficiently conserved for molecular analyses [33] (Table 2).

All of these studies demonstrate several benefits of using psoralen as an inactivating agent. As both nucleic acid and surface proteins integrity are unaffected, this method has applications for immunofluorescence assays, flow cytometry, Sanger sequencing, RT-qPCR and other molecular biology analyses [22,33,38].

### 2.4. UV Radiation

UV radiation (UVC and UVB) causes inactivation through damage of nucleic acids and proteins, either by direct or indirect pathways. The formation of crosslinks between pyrimidines and between nucleic acids and proteins inhibits viral transcription [1,27,55,56,79,80,81]. In addition, it has been demonstrated that UV-radiation of bacteriophage MS2 (used as surrogate for enteric viruses) causes alterations in the capsid protein. This damage prevents the viral genome release into the host cell [1]. UV radiation acts on both viral DNA and RNA [27] but is likely most effective against viruses without genome repair mechanisms (i.e., single-stranded RNA viruses) [1]. UV causes selective protein damage and a small number of adaptative mutations might be enough for the virus to develop resistance against UV [1]. The number of particles in a given volume also affects inactivation, as it reflects the probability that a photon/nucleotide interaction will occur [27]. In 2004, SARS-CoV was shown to be particularly sensitive to UV-light [77]. In a recent study complete inactivation of SARS-CoV-2 was achieved after exposure to a UV (254 nm) energy level of 0.04 J/cm^2^ or higher [72].

A. A. Kraus et al., 2005 examined the ability of UV light to inactivate Hantaan orthohantavirus (HTNV) [9]. HTNV-infected Vero E6 cells were exposed to UV-radiation for 1, 3 or 5 min at 312 nm (corresponding to 0.5, 1.4, and 2.4 J/cm^2^) or to a chemical agent (acetone, methanol or paraformaldehyde). UV-irradiation fully inactivated HTNV after 3 min. Incubation in absolute methanol for 8 min, 1% paraformaldehyde for 20 min, or acetone/methanol (1:1) for 10 min also successfully inactivated the virus [9]. The authors proposed that all Hantaviruses could be inactivated by these methods [9] (Table 2).

### 2.5. Methanol and Acetone

Methanol and acetone are both lipid solvents that cause destruction of the virion envelope through denaturation or coagulation of proteins [9]. The ability to perform molecular assays, however, is not affected. Fixation using 100% methanol has been shown to effectively inactivate SARS-Cov-2 [72].

### 2.6. Formaldehyde

Formaldehyde and paraformaldehyde (the polymeric form of formaldehyde) react with nucleic acids (both RNA and DNA) to form crosslinks between nucleic acids and proteins and between proteins [20,21]. This process is mediated by the aldehyde group (-CHO) that forms methylene bridges (-CH2-) to protein components as well as modifications related to Schiff bases and methyl groups [19,23,24]. Formaldehyde preserves cell structures, is a very common fixative and allows for subsequent electron microscopy, immunofluorescence assays, and flow cytometry [19,22]. Fixation of SARS-CoV-2-infected cells with 4% paraformaldehyde has been shown to effectively inactivate the virus [72,73].

## 3. Serology

### 3.1. Thermal Inactivation

Thermal inactivation can affect serological tests at temperatures exceeding 56 °C [4]. For example, neutralization tests (i.e., plaque reduction neutralizing test (PRNT)), based on the stability of immunoglobin, can become unreliable if the sample is inactivated at temperatures exceeding 56 °C due to protein gelatinization or denaturation [4,5]. Therefore, heat inactivation at temperatures greater than 56 °C is generally discouraged [6]. Rather than increasing the temperature, it may be advisable to increase the heat exposure time [6]. Many protocols for PRNT require the samples to be inactivated at 56 °C for at least 30 min prior to incubation with virus [6].

The application of this protocol to flaviviruses is supported by several studies and is based on the structural similarity of these viruses [5,82]. The thermal stability of Alkhurma Hemorrhagic fever virus was examined and compared with reports of thermal stability of other flaviviruses [74]. The virus was exposed to different temperatures (45, 50, 56 and 60 °C) in 10 min intervals up to a total of 60 min at each temperature setting. The results showed that inactivation follows a linear curve and the rate is directly proportional to the concentration of the virus. Infectivity is completely lost after heating at 56 °C for 30 min, or at 60 °C for 3 min [74]. The pattern for heat inactivation was consistent with the results reported for other flaviviruses [82,83].

Heat inactivation of viruses with a higher thermostability than *Flaviviridae*, such as *Togaviridae*, however, is challenging [82,84]. Chikungunyavirus (CHIKV), for instance, requires 120 min at 56 °C to achieve complete inactivation [84]. This study is in accordance with a recent study [6] which analyzed the thermostability of the *Togaviridae* Ross River virus (RRV), Barmah forest virus (BFV), and O’nyong-nyong virus (ONNV) [6]. While BFV was found to be effectively inactivated by the heat treatment procedure set by WHO (56 °C for 30 min), ONNV and RRV were found to be more heat stable. ONNV was inactivated after 40 min at 56 °C, while RRV, similarly to the reports on CHIKV, required greater than 120 min of incubation at 56 °C to become fully inactivated. For RRV no infectious virions were detected after 140 min at 56 °C [6].

WHO guidelines for BSL 3–4 pathogens suggest the combination of inactivation methods, such as detergent and heat, for serologic testing [8,15,26]. For ELISA-based serological determinations, WHO recommends inactivation using 0.2% sodium dodecyl sulphate (SDS), 0.1% Tween 20 and heat treatment at 60 °C for 15 min [2,8,13,14]. The combination of Tween-20 and heat has been demonstrated to be as efficient against Zaire Ebola virus (ZEBOV) [14] and LASV [13]. Human serum was spiked with a high concentration of ZEBOV (107 TCID50/mL) and incubated for 1 h with 0.5% Tween-20 at 56 °C [14]. Full inactivation was confirmed by a TCID50 assay. In addition, whole-blood samples from patients who either had advanced infection, were recovering from infection or had suspected infection were subject to treatment and the antibody quality was assessed by ELISA [14]. IgG and IgM antibodies showed no reduced quality after exposure to the inactivating agents [14]. This simple approach can have practical applicability in the field setting [14].

### 3.2. Tween-20 and Tween-80

Tween-20 and Tween-80 are mild, relatively non-toxic detergents that do not denature proteins [10,11,16]. Used alone, they are useful for isolation and analysis of membrane proteins. Enveloped viruses are effectively inactivated by disruption of the lipid-based envelope [39]. Antigenicity and immunogenicity of the inactivated virus is affected, but in low dose and/or in combination with other inactivating agents, antibodies are often sufficiently preserved for serologic assays such as ELISA [14,15]. When combined with thermal inactivation, the balance between temperature, concentration and time must be evaluated carefully and likely determined for each species separately [14]. SARS-CoV-2, however, has been shown to retain infectivity after incubation with 0.5% Tween-20 for 30 min at room temperature [72]. This method allows for other biochemical analysis that may be valuable for patient follow up, such as serum glucose, creatinine and electrolytes [11,13].

### 3.3. Gamma Irradiation

Gamma irradiation can also be used to inactivate viruses, but because of multiple influencing factors, such as solute protein content, virus concentration, temperature and possibly the air volume (which reflects the amount of oxygen free radicals produced in the reaction), it requires confirmatory tests of inactivation [63,85,86]. Some studies identify the viral genome size as an influencing factor, but more recent studies have identified exceptions to this trend [85,87]. LASV and other *Arenaviridae* have been shown to require higher doses of radiation for complete inactivation, than for example *Filoviridae* [63]. Gamma-irradiation induces inactivation through damage to nucleic acids and proteins via direct and indirect pathways [85]. The damaged nucleic acids are unable to replicate, leading to inactivation of the virus [59,60,61]. The inactivation curves with gamma are almost exclusively logarithmic. The sensitivity of microbes to gamma irradiation is estimated by appointing D10 values, which is described as the dosage of radiation required to reduce infectivity of a sample by 90% or one log_10_ [85]. Inactivation by gamma irradiation does not significantly alter the biological activity of proteins or the structural integrity of the virion. This enables conductance of biochemical tests and assays requiring preserved antigenicity, such as ELISA and indirect IFA [63]. Susceptibility to gamma irradiation has been assessed for several viruses and it has been suggested that closely related viruses have similar sensitivities to gamma irradiation [63,85]. Gamma-irradiation therefore should be applicable for a majority, if not all, BSL-4 pathogens as they generally share characteristics in being enveloped, single stranded RNA viruses [85]. SARS-CoV-2 has been shown to require irradiation doses as high as 1.0 Mrad to be completely inactivated [73].

### 3.4. Psoralens

Psoralens are photo active agents causing cross-linking of nucleic acid pyrimidine residues upon exposure to UV-A [22,33,38]. As described in section “molecular biology”, psoralens have been shown to effectively inactivate viruses from several different families with inherently different characteristics [22,38]. But psoralens are also suitable for serologic analysis. Psoralens freely penetrate the phospholipid bilayers without affecting viral integrity, thus preserving surface epitopes that mediate antigenicity [22,38]. Psoralens are effective towards *Toga-*, *Flavi-*, *Fili-*, *Peribunya-*, *Arena-*, *Nairo-*, *Phenui-*, and *Orthomyxoviridae*. Antigenicity and immunogenicity were specifically tested for DENV, LASV and Crimean-Congo hemorrhagic fever orthonairovirus (CCHFV) representing *Flaviviridae*, *Arenaviridae* and Nairoviridae families respectively [33]. Psoralen-inactivated DENV was shown to bind to five monoclonal antibodies directed to DENV envelope protein and to elicit an effective T-cell response in mice vaccinated with the virus. Immunogenicity against psoralen-inactivated LASV and CCHFV was tested by infecting rabbits with either inactivated virus or live virus and confirming the production of LASV or CCHFV-specific antibodies respectively [33], indicating that antibody structures are sufficiently preserved for the conduction of serological studies [22,33,38].

### 3.5. 1.5-Iodonaphtyl Azide (INA)

1.5-iodonaphtyl azide (INA) is another photoactive chemical known for its ability to inactivate enveloped viruses without affecting surface epitopes. It has primarily attracted attention for vaccine development (see section on vaccine development), but is equally useful where serological studies are required [22].

### 3.6. Aromatic Disulfides

Aromatic disulfides (Ph-S-S-Ph) and diathianes (ex 1,2-dithiane-4,5-diol) represent a different approach towards virus inactivation (Table 1). They are reactive against the zinc finger motifs of proteins in certain viruses [64,65]. Rice et al., demonstrated in 1996 that the agents 2,2′-dithiobis-[benzamide] and benzisothiazolone inhibit HIV-1 replication by reacting with the nucleocapsid protein NCp7, causing Zn ejection from the protein [88,89,90]. Subsequently, several effective aromatic disulfides have been identified [88]. Other viruses containing a Zn-binding domain (such as *Arenaviridae*) can be inactivated in a similar manner [64,65,91]. The compounds diffuse across the lipid bilayers of the viral envelope, leaving conformational envelopes of proteins intact [64,90]. Antigenic and immunogenic properties of these surface glycoproteins are likely preserved [65,92].

The aromatic disulfide NSC20625 has been described as the most effective disulfide agent [13,64]. The efficiency of NSC20625 against Argentinian mammarenavirus (AHFV) (previously Junin virus) was compared with a diathane NSC624152 [64]. Both caused an immediate decrease in virus infectious titers. NSC20625 was, however, considerably more effective. After exposure to 20 µM NSC20625, the virus was inactivated by 92–95% following a 15–20 min incubation period and by 99.0–99.9% after 90 min [64]. However, no inactivation occurred at 4 °C [64]. In a subsequent study, incubation with 40 µM of NSC20625 for 90 min eliminated the ability of AHFV to form plaques in Vero cells.

Electron microscopy and SDS-PAGE analyses demonstrated that the morphology and function of AHFV surface proteins GP1 and GP2 is preserved (i.e., cell receptor binding/fusion between viral and cellular membrane) [65]. The inactivated virus retains the capacity to bind and become internalized into the host cell, but is contained within endosomal vesicles, preventingfurther uncoating and release of nucleocapsid into the cytoplasm. Consequently, RNA replication, RNA transcription, and protein synthesis are inhibited [64,65,90]. Translation of viral genome into protein was assessed by immunofluorescence detection. No signal was observed after 48 h [64].

Subsequently, immunoprecipitation tests were performed using hyperimmune anti-AHFV serum and demonstrated antibody binding with similar avidity to both inactivated and infectious virus. This suggests that antigenicity and immunogenicity are not altered. The method is thus a promising approach for serological analysis of *Arenaviridae*.

### 3.7. Hydrogen Peroxide (H_2_O_2_)

Hydrogen peroxide (H_2_O_2_) is a strong oxidizing agent that causes genomic damage thus impeding the ability of the virus to replicate. It is effective against both RNA and DNA viruses [25]. It has been shown to preserve antigenic structures capable of inducing both humoral and cellular responses [25,48,49]. Therefore, H_2_O_2_ has attracted attention for serologic analysis and in recent years additionally as an alternative for vaccine development (see section on vaccine development) [25,49,50].

## 4. Vaccine Development

Vaccine development sets very specific demands on virus inactivation. The structure of the viral proteins must be sufficiently preserved to induce an effective host immune response. Specifically, the ability to elicit a neutralizing antibody response is often essential [22,23,50]. Simultaneously, absolute sterility and inability of virus replication must be assured. Since the 1920s, formaldehyde and Beta-propiolactone (BPL) have been the most commonly used inactivating methods for vaccine deployment and until 2020 there had been six licensed viral vaccines that are inactivated with either formaldehyde or BPL [22,23,33,93,94]. Formalin is formaldehyde (gas) dissolved in water and contains 37–40% formaldehyde and 60–63% water (by weight) [32], i.e., 37% *w*/*v* formaldehyde (13.3 M) equals 100% formalin [33] (Table 1). (The terms are used interchangeably and here referenced as put by original authors). Formaldehyde is used in the generation of inactivated virus vaccines against Poliovirus, Hepatitis A virus, Japanese encephalitis virus and tick-borne encephalitis virus [23]. BPL is used for the inactivation of rabies and influenza virus vaccines and more recently in the production of whole-virus vaccines against SARS-CoV-2 (e.g., BBIBP-CorV, Sinopharm; CoronaVac, Sinovac) [23,95]. However, the relevance of these inactivation techniques has increasingly been questioned as safer and better alternatives have become available. Research using psoralens as inactivation method in production of a whole-virus SARS-CoV-2 vaccine has, for instance, showed promising results [96]. Historically, residual infectivity has occurred in formalin-inactivated viruses, which has led to outbreaks upon vaccine distribution of for example measles, Venezuelan equine encephalitis virus, foot-and-mouth disease and Poliovirus vaccines [23,97,98,99].

Inactivated vaccines generally have a higher safety profile than live-attenuated vaccines, and reversion into active disease is very unlikely. They are also less reactogenic, but are associated with a lower immunogenicity, requiring several doses to achieve effective immunity [23]. This is exemplified by the two vaccines that have been developed against Venezuelan equine encephalitis virus (VEEV), TC-83 (live-attenuated and prepared from chicken embryo cells) and C-84 (formalin inactivated). TC-83 (single dose) and C-84 (3 doses elicit similar neutralizing antibodies, but C-84 with less discomfort [100]. C-84 induces high titers of cross-reactive antibodies in non-immune subjects after one primary and two booster vaccinations. The current equine vaccine against VEE relies on a two point attenuating mutation and a concern has been that reversion could generate a VEE epidemic involving large number of human cases [100,101].

### 4.1. Formaldehyde and Beta-Propiolactone (BPL)

Formaldehyde and Beta-propiolactone (BPL are categorized as oncogenic substances by the International Agency for Research on Cancer (IARC) under group 2A and 2B respectively [102,103]. In addition, these agents affect surface epitopes, rendering the inactivated virus less immunogenic [23,60]. BPL is expensive, requires stringent safety guidelines and has been known to trigger adverse immune reactions [49,104,105]. Elimination of BPL from samples through hydrolysis requires an extra step that is time consuming (usually 2 h). However, once hydrolyzed, BPL partitions into lactate and betahydroxypropionic acid and these degradation products have not shown any cell toxicity [23,53,106,107,108]. The inactivating process for these agents is also considerably longer than other available techniques [22,33,93,94]. For formaldehyde, the initial binding to protein is completed in 24 h, but the formation of methylene bridges takes much longer time [19,24]. Complete virus inactivation often requires as much as 2–3 weeks of treatment [22,25]. In the generation of vaccine against Japanese encephalitis virus (*Flaviviridae*) for instance, inactivation requires 50–60 days at 4 °C with formaldehyde concentration of 1/2000 [109]. The time required may be reduced by increasing concentration and temperature, but this influences immunogenicity negatively due to destruction of important epitopes [23,110]. Though inactivation by BPL is less time consuming, it still often requires 24 h’ incubation for complete inactivation [49]. For vaccine generation parameters are generally 18–24 h at 4 °C with BPL concentration 0.1–0.25%, but may vary depending on pathogen [111].

### 4.2. 1.5-Iodonaphtyl Azide (INA)

1.5-iodonaphtyl azide (INA) is a photoactive hydrophobic azide molecule that has recently attracted increased interest for vaccine application (Table 1). Upon exposure to UV-A, the compound produces covalent bonds to viral proteins in the hydrophobic domains of the bio membranes, leading to inactivation [35,112]. The preservation of the ectodomain, including surface epitopes, has marked INA as an interesting inactivating agent for vaccine development [22]. One advantage of INA over other more toxic chemical agents is that it rapidly partitions into lipid bilayers and therefore, is virtually absent in aqueous medium [35]. Any residual INA in the aqueous medium can be inactivated by glutathione [35]. INA has previously been used as an inactivating agent for Venezuelan equine encephalitis virus, Influenza virus, Ebola virus, Vaccinia virus, Pixuna virus, Macaca mulatta polyomavirus 1 (previously Simian virus 40) and Human immunodeficiency virus [113]. It is also effective on Zaire Ebola virus (ZEBOV) [35]. 100 µM of INA was incubated with 2.106 pfu/mL of ZEBOV in PBS for 30 min at room temperature. Residual INA was neutralized using glutathione (20 mM). The sample was then exposed to a UV-dose of 10 mW/cm^2^ for 10 min, using 310 nm UV light and was completely inactivated. [35]. Immunogenicity was demonstrated by vaccinating mice using inactivated virus (once or twice). 4 weeks post-immunization, mice were challenged with a lethal dose of mouse adapted-ZEBOV. All the mice survived and showed no sign of disease [35]. Antibody titers in these animals were measured and confirmed to be once to twice as high as before the challenge. EBOV-specific antibodies were determined by ELISA using whole inactivated virus as antigen. Presence of virus in mice blood was assessed by real-time PCR and confirmed to be negative [35]. The morphology of inactivated virus was examined with electron microscopy and showed to be indistinguishable from non-treated virus. Preservation of surface epitopes were further studied by virus-capture assay. It showed comparable efficiency in the capture of live versus INA-inactivated ZEBOV [35]. T-cell responses in the infected mice were also examined and showed a CD8+ T-cell response to EBOV-specific epitopes [35].

### 4.3. A Comparative Study on the Effect of Formaldehyde (Traditionally Used) and the Photochemical Agents INA and AMT

A comparative study on the effect of formaldehyde (traditionally used) and the photochemical agents INA and AMT on antigenicity and immunogenicity of inactivated DENV (for more details on AMT see section molecular biology, psoralen) [22]. DENV-2 was inactivated by either preparations of 0.02% formalin (pH 7.4) for 120 h at 22 °C, 50 µM INA or 2 mM glutathione with exposure to UV (~365 nm for 5 min, ~145 mW/cm^2^) or 35 µM AMT with exposure to UV (~365 nm, 2 min, ~145 mW/cm^2^) [22]. Virus inactivation was confirmed by infection of Vero cells with inactivated virus for 7 days and subsequently assessed by IFA for dengue virus E and prM proteins, using specific antibodies [22]. Direct plaque assay was also performed. The results showed that all three agents effectively inactivate the virus. While formaldehyde required 5 days to achieve full inactivation, INA required only 5 min and AMT 3 min [22]. Antigenicity was assessed by examining the ability of a panel of 5 monoclonal antibodies to bind to the DENV-2 E protein. Interestingly, INA and formaldehyde-inactivated virus exhibits a decreased antibody binding capacity by 30–60%, while AMT-inactivated virus bound all five antibodies equally well as untreated virus [22]. Mice vaccinated with 1 µg INA- or AMT-inactivated virus elicited an antibody response equivalent to live virus. When formaldehyde inactivated virus was increased to a 10 µg dose, the antibody response was similar to the antibody response induced by the two other agents [22]. High antibody titers were detectable through day 150 for all three agents. The T-cell responses in vaccinated mice was measured by IFN-γ ELISPOT assay against capsid (C), envelope (E) and non-structural protein NS-1. INA and formaldehyde elicited significantly lower T-cell responses than AMT inactivated virus [22].

This study supports previous studies showing AMT is an effective inactivating agent that can be used on highly pathogenic viruses of several virus groups (both DNA and RNA viruses) and provide a platform for viral vaccine development [33]. The results on INA are more inconclusive [114]. Previous studies on HIV and influenza virus inactivated by INA have shown unaffected binding of monoclonal antibodies, relative to untreated virus [36,115]. In contrast, antigenicity decreases upon inactivation of CHIKV and DENV by INA [22,113]. It has been suggested that the decreased immunogenicity of INA may be prevented by using free radical scavengers in the reaction, as the free radicals produced by UV irradiation may damage surface epitopes [22].

### 4.4. Beta-Propiolactone (BPL) and Other Alkylating Agents

Beta-propiolactone (BPL) is the second most commonly used inactivation method for vaccine development after formaldehyde [23,26,42]. It is an alkylating agent that modifies the structure of nucleic acids forming crosslinks between DNA and proteins, and between DNA strands in the double helix (Table 1) [52]. Transcription is impeded which leads to inactivation of both DNA and RNA viruses [52]. Though BPL primarily causes alkylation of guanosine (which is consequently misread by polymerases as adenosine), BPLS has also been shown to react with nine different amino acids which may affect the stability of protein structures. The conformations and functions of important epitopes responsible for immunogenicity are affected to varying degrees [23,54,107]. The hemagglutinin fusion peptide of influenza H1N1 is targeted by BPL, inhibiting its fusion to the host cell [54]. However, susceptibility may vary between different strains. H1N1 was, for instance, found to be significantly less sensitive to BPL than H3N2 [53]. Hemagglutination assay (HA) titers were used to determine virus susceptibility. BPL concentrations ≥0.05% showed a decrease in the HA titer of H3N2 strains while H1N1 virus strains were not or only slightly affected [53]. Dynamic light scattering analyses and western blot analyses demonstrated that BPL concentration influences virion size and integrity. At low BPL concentration (0.02%), the virion diameter was unaffected and there was no decrease in HA titers, suggesting that virion structure and antigenicity was preserved [53]. In contrast, 0.1% BPL concentration caused structural perturbations.

Consistent with these findings, Jonges et al., 2010, also reported a negative effect on both hemagglutinin and neuraminidase functions of the H3N2 virus tested after inactivation by 0.1% BPL [60,110]. While it has been acknowledged that different subtypes of influenza viruses require different physical and chemical conditions for inactivation [110], the reason for this variable effect of BPL remains unclear. One hypothesis is that BPL induces a higher degree of alkylation towards polar than non-polar amino acids and thus a higher degree of modification to the M1 protein in H3N2 than in H1N1 [53,54]. The M1 protein is the most abundant structural protein in the virion and excessive alkylation to it may lead to instability of the virion structure [53]. In addition, BPL may affect hemagglutinin titers indirectly by decreasing the pH during hydrolysis [53,60,116]. The effect on protein stability, however, was shown to have limited effect at low BPL concentrations (0.02%) [53].

In another study, nine avian influenza (AI) viruses were tested for susceptibility to BPL, ether and formaldehyde: four highly pathogenic avian influenza (HPAI) H5N1 viruses, two reverse genetically modified H5N1 and three low pathogenic avian influenza (LPAI) viruses (H9N2, H4N6, H11N1) [110]. Virus inactivation was determined by infecting embryonated chicken eggs with inactivated virus. Hemagglutination assay (HA) titers were used to determine virus titers. Hemagglutination inhibition assay (HI) was performed to determine immunogenicity.

The results showed that all the tested viruses were inactivated after incubation in 0.1% BPL (for 16 h) at 4 °C and in 0.04% formalin (for 16 h) at 37 °C respectively. The concentration of formalin could be decreased to 0.02% if incubation time was increased to 48 h at 37 °C. Correspondingly, increasing the concentration reduced inactivation time, but adversely also altered the HA-titers. The tests using ether were variable, and non-effective against three of the HPAI viruses tested. When combined with Tween-20 (0.2 or 0.5%) all the tested viruses exposed to ether were effectively inactivated. There was no significant drop in HA-titers after treatment with BPL, ether or formalin, indicating preserved antigenicity of the inactivated virus. The HI-assay demonstrated antibody-titers at a significant level indicating that the virus had preserved immunogenicity [110].

The alkylating agents formaldehyde and BPL were also investigated for antigenic and immunogenic properties upon inactivation of West Nile virus (WNV) [117]. The virus was grown in Swiss albino mice and BHK-21 cells to a high titer, then inactivated using 0.1% BPL and 0.2% formaldehyde respectively. Successful inactivation was shown by both the absence of cytopathic effect (CPE), and by PCR, with no trace of viral nucleic acid amplification detected. Full inactivation was observed after 48 h (0.1% BPL) and after 120 h (0.2% formaldehyde). Antigenicity was assessed using HA assay at pH 6.2, 6.4 and 6.6. Titers were similar for both agents at pH 6.6 and the humoral immune response measured showed no significant difference. At pH 6.2 and 6.4, HA titers of BPL treated antigen were considerably lower, while formaldehyde treated antigen titers were unaffected. Previous studies have shown that BPL-inactivated virus was significantly more antigenic than formaldehyde-inactivated virus [116]. In this study, the neutralizing antibody response to antigen was similar for both methods. However, the effect on cell-mediated immune system was not assessed.

These studies demonstrate that the preservation of antigenicity after inactivation by BPL is variable and relies on factors such as virus strain, BPL concentration, incubation time and pH.

### 4.5. H_2_O_2_

The ability of H_2_O_2_ to replace formaldehyde and BPL as an inactivating agent for vaccine generation against Yellow Fever virus (YFV) and West Nile virus (WNV), was examined by Amanna et al. [25]. Hydrogen peroxide (H_2_O_2_) is a strong oxidizing agent that causes genomic damage by attacking the carbon double bonds in the nucleosides or abstract hydrogen atoms [25,47]. This leads to single- or double-strand breaks that destroys the ability of the virus to replicate. It is effective against both RNA and DNA viruses [25]. H_2_O_2_ has the benefit of being environmentally safe as it is decomposed into oxygen and water and does not require complicated purification processes to be removed from vaccine preparations. Additionally, it has also been shown to preserve antigenic structures capable of inducing both humoral and cellular responses [25,48,49]. The same study also showed that H_2_O_2_ was an extremely effective inactivating agent, causing 6-log_10_ reduction in titer in less than 2 h for both RNA and DNA viruses, after exposure to 3% aqueous solution of H_2_O_2_.

The authors also found that H_2_O_2_ causes minimal damage to YFV viral epitopes. They compared inactivation of YFV by H_2_O_2_, BPL or formaldehyde. After coating to ELISA plates, inactivated virus was probed with YFV-immune serum from mice and the ability to bind to antibodies was assessed. 87–98% of the maximum antibody binding response was observed after 2 h of treatment with H_2_O_2_, while BPL- and formaldehyde-treated YFV had significantly reduced antigenicity [25]. To assess the efficiency of H_2_O_2_ as an inactivating agent for vaccine development, mice were then vaccinated with vaccinia virus (VV) inactivated by either H_2_O_2_ or formaldehyde. A new injection was done on day 28 to boost the immune response. H_2_O_2_-inactivated virus induced significantly higher levels of virus-specific neutralizing antibody than the virus inactivated by formaldehyde. Mice immunized with H_2_O_2_-inactivated VV were then challenged with a lethal dose of VV and 100% of the vaccinated animals survived [25]. H_2_O_2_ inactivated WNV also induced antibody production in vaccinated mice exceeding that which was observed in human subjects 1 year after natural WNV infection. The vaccinated mice survived a lethal challenge with no sign of disease or weight loss. Full protective immunity was observed >280 days after vaccination, indicating the induction of long-term protective immunity [25]. Furthermore, CD8+ T cell immunity response was assessed by vaccinating mice with purified H_2_O_2_- inactivated lymphocytic choriomeningitis virus (LCMV). 8 days post-vaccination, LCMV-specific CD8+ T cells reached a similar level to recombinant VV expressing LCMV nucleoprotein (NP)21 or glycoprotein (GP)22. The mice were challenged with 2 × 10^5^ PFU of LCMV at 28 days after vaccination with H_2_O_2_-inactivated LCMV. The mice showed a >40-fold increase in LCMV -specific CD8+ T cells within 4 days. Induced T cells were cytolytic, polyfunctional and could provide protective immunity against chronic viral infection [25].

A study performed by J. L. Dembinski et al., 2014 demonstrates the applicability of H_2_O_2_ on influenza virus with focus on antigenic and immunogenic properties [48]. Three strains were tested: seasonal H1N1, pandemic H1N1 (2009), and seasonal H3N2. The viruses were inactivated using either 1% or 3% H_2_O_2_ solution in PBS for 2 h in room temperature. The results showed that 2 h incubation in a concentration as low as 1% of H_2_O_2_ was sufficient to inhibit replication. Evidence of inhibition was assessed by infecting MDCK cells with inactivated virus and confirming there was no CPE. The virus replication was measured using PCR and showed no sign of replication [48]. Antigenicity was tested using ELISpot. Peripheral blood mononuclear cells (PBMCs) were stimulated by either untreated influenza virus or virus inactivated by H_2_O_2_. There was no difference in the production of INF-γ. The cytokine response was also tested using a microbead-based immunoassay. PBMCs were stimulated and the response measured by analyzing levels of IL-2, IL-10 and TNFα. The cytokine response for cells stimulated by untreated influenza virus was comparable with those stimulated by inactivated virus [48]. The humoral response was also assessed using ELISA by evaluating influenza specific IgG response [48].

Taken together these studies demonstrate that H_2_O_2_ is an effective inactivating agent that can be used to provide a vaccine platform for several virus strains [25,48,49].

### 4.6. Gamma Irradiation

Gamma irradiation has also been suggested as a possible inactivation method for vaccine production. Advantages include the capacity to inactivate large volumes with the virus contained in closed containers or even in a frozen state [118]. Moreover, the purification steps necessary after inactivation with chemical substances can be avoided [118]. However, there remain concerns regarding biosafety and the risk of residual infectivity. The kinetics of inactivation is logarithmic (D10 value) and therefore theoretically 100% sterility cannot be guaranteed [85]. To reduce the oxidative stress caused by irradiation on immunogenic neutralizing epitopes, the addition of an antioxidant molecule has been suggested in vaccine development. The addition of Mn^2+^-peptide antioxidants derived from bacteria Deinococcus radiodurans have been considered in the production of polio vaccines [62]. Though experimental vaccines (against influenza virus and Venezuelan equine encephalitis virus (VEEV) among others) have been developed, no gamma-inactivated vaccine exists today [118,119]. Initial studies indicate that gamma-inactivated pathogens retain their ability to induce T-cell immunity and cross-reactive T-cells, a characteristic highly desirable for vaccine development, which could help facilitate development of a universal influenza vaccine [118]. Other studies indicate reduced antigenicity of inactivated virus due to effects on essential surface epitopes [23,118,119]. Further research will thus be required, but gamma irradiation remains a possible candidate for generation of inactivated whole virus vaccines.

## 5. Conclusions

The purpose of the article has been to identify available inactivating agents and demonstrate their applicability and limitations for specific viruses and settings, with a focus on highly pathogenic viruses. For strict nucleic acid analysis, guanidine based chaotropic salts remain suitable for most highly pathogenic viruses, but may require some caution upon selection [15,26,32,70].

As complete inactivation for highly pathogenic viruses is so critical, several studies suggest the combination of inactivating techniques including at least two different modes of action, e.g., one physical and one chemical [8,15,26]. For serological assays a combination of 0.2% SDS, 0.1% Tween-20 and heat (60 °C for 15 min) has been suggested (WHO) or, as have been shown for inactivation of EBOV, Tween-20 and heat (56 °C for 1 h) in combination provides a method which also preservesthe viral structure [8,14].

Psoralen and 1.5-iodonaphtyl azide (INA) are alternatives for molecular biology analyses, but these have additionally, been shown to preserve virion structure sufficiently to allow for serologic testing and possibly even vaccine development [33,35,96]. Heat, UV-light radiation and gamma irradiation are established inactivation methods against multiple virus species, but each has its limitations that suggest that their application often is best achieved in combination with other inactivating methods. Their addition to inactivating regimes allows for lower concentration of subsequent denaturing agents, thus reducing the effect on virus structure. These combination regimes are often compatible with serologic assays including ELISA, biochemical and immunological studies [2,4,5,8,14,15,85].

Gamma irradiation has advantages as an inactivation technique including its high penetrative power and the lack of purification steps needed after inactivation with chemical counterparts. However, there are still some concerns regarding biosafety, reduced antigenicity and that the equipment is too bulky and expensive for some settings.

Thermal inactivation is simple, follows first-order kinetics, but is affected by multiple factors such as serum protein content, pH, density of the solution, the presence of organic compounds and salts etc. and offers less reliable inactivation [3,8]. Similar difficulties present with UV-light inactivation, including the risk of acquired resistance [1].

Diathanes and disulfide-based compounds represent a totally different approach to virus inactivation. They are specifically targeted to cause alterations in zinc finger motifs within viral proteins, without affecting the integrity of the virus structure. The application, however, is limited to viruses containing such aprotein, such as *Arenaviridae*, HIV and respiratory syncytial virus [65,120]. Other viruses where these agents have application remain to be discovered.

In conclusion, photoactive compounds and hydrogen peroxide (H_2_O_2_) stand out as particularly interesting agents for virus inactivation. They are fast acting, have broad application and an appealing safety profile. There are several photoactive compounds of which only psoralen and INA are mentioned here. Other alternatives not discussed here include methylene blue and riboflavin (Table 1 and Appendix A). Though they have an excellent safety profile and are frequently used in e.g., transfusion medicine, there is insufficient literature to support their application on highly pathogenic viruses [39,42,43,44,45,46,121,122,123,124]. Psoralens stand out as the most potent photoactive chemicals, having broad applicability and the potential to inactivate a diverse set of virus families [22,33].

Hydrogen peroxide (H_2_O_2_) is an extremely potent agent that dissolves into biologically safe compounds, while preserving antigenicity and immunogenicity on a number of different virus families [25,48,50]. J. L. Dembinski et al., 2014, for instance, demonstrate that H_2_O_2_ may be applied to form a vaccine platform against Orthomyxoviridae [48] and further support is attained by ex the studies of Amanna et al., 2012 [25].

This article does not claim to be conclusive; the handling and inactivation of highly pathogenic viruses require careful considerations, as pointed out in this article. Our intention has been to create an overview over available inactivation techniques to aid in the selection and appropriate application.

## Figures and Tables

**Table 1 pathogens-11-00271-t001:** Inactivating agents with correlation to application: molecular biology analysis, serology or vaccine development.

	Structure	Characteristic/Mode of Action	Application (as Described by the Referenced Article)
HEAT		Acts on viral capsid and envelope. Causes structural alterations on viral proteins that disrupt the specific structures necessary to recognize and bind to host cells, thereby inhibiting replication [1]. Viral genomic RNA becomes denaturated at temperatures higher than 60 °C [2,3]. Immunoglobin denatures at temperatures higher than 56 °C [4,5].	Molecular biology and serology. Highly variable results for viruses, even within same virus family [6,7]. Inactivation protocol must be designed for each species separately. WHO recommends that heat is used in combination with a denaturing chemical agents for BSL category 3–4 viruses [8].
SOLVENTS Alcohol/acetone		Lipid solvents. Inactivation by destruction of envelope [9].	Molecular biology, e.g., immunofluorescence assay (IFA) Hantaan orthohantavirus (HTNV) [9]. Fixation of cells with methanol allows for immunostaining [9].
IONIC DETERGENTS	Composed of charged head group (cation or anion) and a hydrophobic chain [10,11].	Breaks down protein and lipid associations, denatures proteins and other macromolecules, ultimately causing disruption of capsid and membranes [10].	
Sodium dodecyl sulfacte (SDS)		Harsh and effective [10,11]. Solubilizes almost all proteins. It acts by disrupting non-covalent bonds within and between proteins, causing conformational changes of the proteins and loss of their function [12].	Serology, in combination with heat and Triton-X100 (BSL-4 pathogen) [2,8,13,14]
NON-IONIC DETERGENTS	Consist of a hydrophilic headgroup and a hydrophobic tail.	Breaks protein and lipid associations, denatures proteins and other macromolecules, ultimately causing disruption of capsid and membranes [10].	Molecular biology and serology: Useful for isolation and analysis of membrane proteins. Inactivates enveloped viruses by disruption of the lipid-based envelope. Nucleus and nucleic acid are left intact [10]. Affects antigenicity and immunogenicity, but in low dose and/or in combination with other agents used for serological analysis [14,15].
Triton X-100	Derived from polyoxyethylene. Contains an alkylphenyl hydrophobic group [10,11].	Mild surfactants. Breaks protein-lipid and lipid-lipid associations, but not protein-protein interactions. Proteins can be isolated in their native and active form. At low dose, antigenicity is not affected [10,11,16].	Molecular biology and serology: Lyse, fix and permeabilize cells [10,11]. Effective against a wide range of enveloped viruses [17]. Often used in combination regimes with e.g., heat or other denaturing agents [13,15].
Tween-20/ Tween-80	Polysorbant surfactants composed of a fatty acid ester and a long polyoxyethylene chain [10].	Similar to Triton X-100	Molecular biology and serology: Mild and do not affect protein activity or enzymatic assays [11,16].
ALDEHYDES		Causes alkylation of amino- and sulfidryl- groups of proteins and purine bases, leading to stable cross-links that prevent viral replication [18].	
Formaldehyde (CH_2_O)	Gas that rapidly dissolves in water to form methylene hydrate (equally reactive as formaldehyde) [19]. Formalin liquid (polymerization of methylene hydrate (*n* = 2–8)) contains 37–40% formaldehyde and 60–63% water (by weight) [19].	Causes a great diversity of modifications exerted by methylol groups, Schiff bases and methylene bridges. Forms nucleic acid-protein and protein-protein crosslinks [20,21]. Initial binding to proteins is completed in 24 h, but formation of methylene bridges often requires as much as 2–3 weeks of treatment [22,23,24,25].	Serology and vaccine development [26]. Common fixative, used for e.g., IFA, flow cytometry and electron microscopy. Preserves cell structures and allows for staining of cell surface antigens and immunoassays. Some reports indicate reduced chemicals and antigenic properties due to conformational changes in the structure of proteins. [22].
Glutaraldehyde (HCO-(CH_2_)_3_-CHO)	Composed of a chain of 3 methylene bridges with an aldehyde group at each end [19].	Considerably more reactive than formaldehyde due to multiple reactive aldehyde groups. Reacts with proteins (minutes to hours) but penetrates tissues slowly. It is less volatile and produces less cross-linking with nucleic acids than formaldehyde [19,27,28,29,30,31].	Molecular biology. Disinfectant or fixative.
Paraformaldehyde (OH(CH_2_O)_n_H)	Polymeric form of formaldehyde (*n* = 8–100), precipitate as solid white powder [19].	Dissolved to form monomeric formaldehyde or methylene hydrate.	Same applications as formaldehyde.
GUANIDINE BASED CHAOTROPIC SALTS (nucleic acid extraction kits), e.g., Trizol LS, AVL buffer, Magna pure lysis/binding buffer (MPLB).	Buffers containing guanidine isothiocyanate in combination with other ingredients (specific for each buffer system) e.g., phenol or Triton X-100 [12,32].	Extremely caustic. Denatures macromolecules such as DNA, RNA and proteins [33,34].	Molecular biology. Extraction and analysis of nucleic acids.
PHOTOACTIVE COMPOUNDS			
1.5-iodonaphtyl azide (INA)	A photoactive hydrophobic azide molecule that is converted to a reactive nitrene radical upon UV-A exposure [35,36].	INA associates with lipid membranes, and upon exposure to UV-light, thenitrene radical which is products interacts with the transmembrane domains of viral proteins, formingcovalent bonds to the hydrophobic domains. The ectodomains, protruding outside the membrane, are left intact [22,36,37].	Molecular biology, serology and vaccine development. The preservation of surface epitopes has marked INA as a promising inactivation agent for vaccine development [22].
Psoralen	Small photo reactive, naturally occurring compounds structurally related to coumarin [38].	Psoralens freely penetrate phospholipid bilayers and intercalate between nucleic acid pyrimidine residues, causing covalent crosslinks upon exposure to UV-A. Inactivation results from inhibition of DNA replication and RNA transcription [22,33,39,40].	Molecular biology, serology and vaccine development [22,33,40,41]. Activity against a broad range of viruses, including enveloped, non-enveloped, DNA and RNA viruses [33]. Inactivation is dose-dependent and species-specific, maybe even strain-specific [33].
Methylene blue (MB)	A thiazine dye [42,43]	In the presence of oxygen and UV exposure, MB binds strongly to DNA (G-C rich regions) and mediates nucleic acid strand breaks [42,43]. The effect is at least partly attributed to the formation of singlet oxygen and the formation of different oxygen products [42,43].	Unclear. MB does not penetrate cells and therefore cannot inactivate intracellular viruses. Is mainly used to inactivate blood products that have been depleted from leukocytes [39]. Effectivity has been recognized against e.g., CHIKV, WNV, DENV [39,42,44].
Riboflavin (vitamin B2)	The molecule is a planar conjugated ring structure with a sugar side chain that confers water solubility. It is an essential human nutrient [B2].	Intercalates between the bases of DNA or RNA and upon exposure to UV-light oxidizes the guanine nucleotide, thereby preventing viral replication [39,45,46].	Unclear. Effective against a broad range of viruses, both enveloped and non-enveloped [43]. Routinely used in transfusion medicine for the inactivation of blood components [39,45,46].
OXIDIZING AGENTS			
Hydrogen peroxide (H_2_O_2_)		Hydroxyl radicals mediate genomic damage by attacking the carbon double bonds in the nucleosides or abstract hydrogen atoms. Single-, or double-strand breaks results, leading to inhibited viral replication [25,47].	Serology and vaccine development [25,48,49,50]. Effectivity against both RNA and DNA viruses [25]. Commonly used as disinfectant and sterilization of different surfaces and surgical tools [25,51].
ALKYLATING AGENTS			
Beta-propiolactone (BPL)		Modifies the structure of nucleic acids inducing nicks in the DNA, crosslinks between DNA strands and between DNA and proteins [52]. Primarily targets guanine bases. The BPL-modified guanine is misread by the polymerase as adenine. For every alkalated guanosine, a GC-AT transition mutation is incorporated. Transcription is inhibited, leading to inactivation of both RNA and DNA viruses [52]. At higher concentrations, BPL causes instability of protein structures which leads to disruption of the virion structure [53,54].	Serology and vaccine development [26,42,54]. The effect on virion structure and antigenicity is dose-dependent [53].
IRRADIATION			
UV-radiation	UVC and UVB	Directly induces crosslinking between adjacent pyrimidines into dimers, and between proteins and nucleic acids [27,55,56,57,58]. Also acts through an indirect pathway by formation of reactive oxygen species that damage nucleic acids and proteins [1,55,56].	Serology. Effect on both viral RNA and DNA [1,27].
Gamma irradiation		Direct pathway: mediated by radiolytic cleavage or crosslinking of genetic material [59,60,61]. Indirect pathway: formation of free radicals from radiolytic cleavage of water (hydroxyl radicals) and oxygen (singlet oxygen). These molecules react with nucleic acids and proteins leaving the virus incapable of replicating [59,60,61,62]. The indirect pathway constitutes two-thirds of the effect on inactivation [59].	Serology including immunoassays, biochemical and immunological studies [62,63].
DISULFIDE BASED COMPOUNDS	Aromatic disulfides (Ph-S-S-Ph) Diathanes (e.g., 1,2-diathane-4,5-diol)	Reactive against the zinc finger motifs of proteins in certain viruses [64,65]. These interal matrix proteins are often essential for genomic replication and nucleocapsid assembly [64,66]. RNA transcription, replication and protein synthesis are inhibited. The compounds diffuse across lipid bilayers of the envelope, leaving conformational envelope proteins intact [64].	Serology and vaccine development [65] Effective against a small selection of viruses containing zinc finger motifs, e.g., *arenaviridae* and HIV [65].
ARGININE	Amino acid, naturally occuring, highly hydrophilic and polar. Contains a protonated guanidium group that forms ionic and hydrogen bonds with proteins and lipids. [67,68].	Acts on both RNA & DNA viruses, but exlusively on enveloped viruses. Most likely interacts with lipid membrane of the virus, but the mechanism of action has not been fully elucidates. Proteins are not damaged by inactivation. Synergistically inactivates at less acidic pH (pH > 4) or lower temperature (30–40 °C) increasing protein stability and yield [67,68].	Application is primarily within therapeutic protein manufacturing. Used for inactivation of for instance influenza virus and herpes simplex virus. Applications in clinical diagnostics remain to be investigated [67,68].

**Table 2 pathogens-11-00271-t002:** Dose-requirements for inactivation of different viruses using 4′-aminomethyl-trioxsalen (AMT) and UV-A light, Schneider et al., 2015 [33].

Family	Virus	AMT [µg/mL]	UV Exposure (min)	365 nm UV-A (µW/cm^2^)	Total Energy Exposure (µW/cm^2^)
*Togaviridae*	VEEV	20	20	1000	2000
*Filoviridae*	EBOV	20	120	1000	120,000
	MARV	20	150	1000	150,000
*Flaviviridae*	DENV	10	40	200	8000
	WNV	ND	ND	ND	2000
	SLEV	ND	ND	ND	1000
	YFV	ND	ND	ND	2000
*Arenaviridae*	LASV	10	150	200	30,000
	AHFV(JUNV *)	20	90	1000	90,000
*Nairoviridae*	CCHFV	10	20	200	4000
*Phenuiviridae*	RVFV	20	90	1000	90,000
Orthomyxoviridae	H1N1p	10	ND	ND	1000
	H1N1p	10	ND	ND	1000
	H3N2	10	ND	ND	2000
	Influenza B	10	ND	ND	2000

Venezuelan equine encephalitis virus (VEEV), Ebola virus (EBOV), *Marburgvirus* (MARV), Dengue Virus (DENV), West Nile Virus (WNV), St. Louis encephalitis virus (SLEV), Yellow fever virus (YFV), Lassa mammarenavirus (LASV), Argentinian mammarenavirus* (Junin virus JUNV), Crimean-Congo hemorrhagic fever orthonairovirus (CCHFV), Rift Valley fever phlebovirus (RVFV). ND = not demonstrated.

## Data Availability

Not applicable.

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
