# Peer review of "Methods of Inactivation of Highly Pathogenic Viruses for Molecular, Serology or Vaccine Development Purposes"

_pathogens, 2022, doi:10.3390/pathogens11020271_

Round 1

Reviewer 1 Report

The review presented a good summary of methods for viral inactivation targeting diagnosis or vaccine development. I guess that this theme may of great interest of Pathogens readers. The manuscript is well written, but there are some minor points that must be addressed.

  • The references may require some update, since from 118 references, only 5 (less than 5 %) were related to the last 5 years, while 31 (~26%) references were related to the last 10 years. [2021 (1), 2020 (0), 2019(1), 2018 (1), 2017 (2), 2016 (5), 2015 (6), 2014 (4), 2013 (5), 2012 (6)]. It is worth to highlight the recent increasing of publications of this theme in the last 2 years due to pandemic situation.
  • There are some studies pointing the use of arginine for inactivation of enveloped virus, I guess that these studies could be also included.

Author Response

Reviewer 1:

The review presented a good summary of methods for viral inactivation targeting diagnosis or vaccine development. I guess that this theme may of great interest of Pathogens readers. The manuscript is well written, but there are some minor points that must be addressed.

Thank you

The references may require some update, since from 118 references, only 5 (less than 5 %) were related to the last 5 years, while 31 (~26%) references were related to the last 10 years. [2021 (1), 2020 (0), 2019(1), 2018 (1), 2017 (2), 2016 (5), 2015 (6), 2014 (4), 2013 (5), 2012 (6)]. It is worth to highlight the recent increasing of publications of this theme in the last 2 years due to pandemic situation.

Thank you for this comment. This has been completed (page 5, lines 105-108 and 127-133; Page 7, lines 186-187, 199-200, 208-210; page 8 line 261-263; page 9 lines 286-287; page 11 lines 371-381)

There are some studies pointing the use of arginine for inactivation of enveloped virus, I guess that these studies could be also included.

Thank you. Arginine inactivation has been added (Table 1 page 2)

Reviewer 2 Report

  1. There are numerous grammatical mistakes throughout
  2. Abstract - there are missing key words such as cells or fixing, I'm not sure a search engine would pick this up.
  3. Table 1 - You mention 'serology' do you mean immunoassay?
  4. Table 1 - Gamma radiation with a Mn-Peptide was shown by Daly, et al PMID: 31999745, you might want to update the ref's
  5. Page 5 - desiccation wasn't mentioned
  6. Page 7 - Are you sure you mean 'thermodynamics' here? 
  7. Page 11 - 'A' is not an appropriate abbreviation

Author Response

Reviewer 2

  1. There are numerous grammatical mistakes throughout

Thanks for this comment. The English has now been corrected by a native English speaker

  1. Abstract - there are missing key words such as cells or fixing, I'm not sure a search engine would pick this up.

Thank you for this comment. “fixing” has been added in the abstract. We didn’t add cells because we consider it not to be specific enough.

  1. Table 1 - You mention 'serology' do you mean immunoassay?

Thank you for interesting remark. We really meant serology (detection of antibodies using virus) but we added immunoassay (detection of antigens using known antibodies) because it is also important

  1. Table 1 - Gamma radiation with a Mn-Peptide was shown by Daly, et al PMID: 31999745, you might want to update the ref's

True. Thank you for this remark, Corrected in Table 1 and also page 15 lines 595-599

  1. Page 5 - desiccation wasn't mentioned.

Thank you for this comment. We decided to do not talk about dessication because it is not a good way to definitively inactivate most of viruses, neither bleach because we didn’t orientate the review to decontamination of surfaces

  1. Page 7 - Are you sure you mean 'thermodynamics' here? 

Corrected page 8 line 233. Thank you for your comment

  1. Page 11 - 'A' is not an appropriate abbreviation

Thank you for this comment. It has been corrected.